

# Genome-wide identification and characterization of the *OFP* gene family in Chinese cabbage (*Brassica rapa* L. ssp. *pekinensis*)

Ruihua Wang[1], Taili Han[2], Jifeng Sun[2], Ligong Xu[2], Jingjing Fan[1], Hui Cao[1] and Chunxiang Liu[1]

[1] Biological and Agricultural College, Weifang University, Weifang, China
[2] Vegetable Research Institute, Weifang Academy of Agricultural Sciences, Weifang, China

## ABSTRACT

Ovate family proteins (OFPs) are a class of proteins with a conserved OVATE domain that contains approximately 70 amino acid residues. OFP proteins are plant-specific transcription factors that participate in regulating plant growth and development and are widely distributed in many plants. Little is known about OFPs in *Brassica rapa* to date. We identified 29 *OFP* genes in *Brassica rapa* and found that they were unevenly distributed on 10 chromosomes. Intron gain events may have occurred during the structural evolution of *BraOFP* paralogues. Syntenic analysis verified *Brassica* genome triplication, and whole genome duplication likely contributed to the expansion of the *OFP* gene family. All *BraOFP* genes had light responsive- and phytohormone-related cis-acting elements. Expression analysis from RNA-Seq data indicated that there were obvious changes in the expression levels of six *OFP* genes in the *Brassica rapa* hybrid, which may contribute to the formation of heterosis. Finally, we found that the paralogous genes had different expression patterns among the hybrid and its parents. These results provide the theoretical basis for the further analysis of the biological functions of *OFP* genes across the *Brassica* species.

## INTRODUCTION

Chinese cabbage (*Brassica rapa* L. ssp. *pekinensis*) is a vegetable belonging to the Brassicaceae family whose commercial quality and value is affected by its leaf traits. Heterosis has been widely used in the breeding of Chinese cabbage, and can effectively increase its yield (*Zhang, 2012*). *Brassica rapa* L. ssp. *pekinensis* var. weichunbai was synthesized by crossing 'BZ07-09' (maternal parent) and 'BD05-272' (paternal parent), resulting in a plant with obvious heterosis in terms of yield, disease resistance, and quality (*Han et al., 2018*). In recent years, researchers have explored the molecular mechanism of heterosis at the level of gene expression using RNA-seq, and the results show that differential gene expression is closely related to the formation of heterosis. The differentially expressed genes in wheat hybrids are involved in biological pathways including photosynthesis and carbon fixation,

Corresponding authors
Ruihua Wang, wfxywrh@wfu.edu.cn
Chunxiang Liu, chunxiang-gliu@126.com

which promote the formation of heterosis, when compared to their parents (*Liu et al., 2018*). The differentially expressed genes in sorghum hybrids are related to the formation of grain yield dominance (*Jaikishan et al., 2019*). Changes in gene expression levels alter specific physiological and biochemical reactions, which lead to trait changes and the formation of dominant phenotypes.

The *OFP* gene family is a class of the plant-specific transcription factor family, which contains a putative bipartite nuclear localization signal and two Von Willebrand factor type C domains used for protein–protein interactions (*Liu et al., 2002*). The first identified *OFP* gene was found to control the fruit shape in tomato as a major quantitative trait locus. A point mutation in this gene leads to the production of a premature stop codon so that the most of conserved OVATE domain is eliminated, resulting in the transition of fruit shape from round to pyriform (*Liu et al., 2002*). *OFP* genes have also been identified and analyzed in multiple species including grape (eight), *Arabidopsis* (19), apple (26), rice (31), and *Zea mays* (45), and they are widely distributed throughout the plant kingdom (*Liu et al., 2014a*; *Yu et al., 2015*; *Wang et al., 2018*; *Li et al., 2019*). The *OFP* genes have been reported to act as transcriptional repressors to regulate multiple biological processes of plant growth and development. For example, in *Arabidopsis thaliana*, AtOFP1and AtOFP4 interact with the transcription factor KNAT7 (KNOX family) to form a functional complex, act as a transcriptional repressor, or regulate secondary cell wall formation (*Li et al., 2011*). AtOFP1 was also found to interact with BLH3 (BELL transcription factor) to regulate the transition from the vegetative phase to the reproductive phase (*Zhang et al., 2016*). AtOFP1, as a transcriptional repressor, also could directly regulate the expression of its target gene, *AtGA20ox1* (a gibberellic acid biosynthetic gene), preventing cell elongation (*Wang et al., 2007*). AtOFP5 interacts with KNAT3 and BLH1 as a negative regulator of the BELL–KNOX TALE complex to participate in early embryo sac development in *Arabidopsis* (*Pagnussat, Yu & Sundaresana, 2007*). In rice, OsOFP2 interacts with KNOX and BELL proteins to modulate the KNOX-BELL complex function, regulating vascular development (*Schmitz et al., 2015*). OsOFP8 (phosphorylated by OsGSK2) is involved in the brassinosteroid signaling pathway in rice (*Yang et al., 2016*). Finally, *OFP* genes are involved in the response to abiotic stresses including drought, cold, and osmotic stresses (*Ma et al., 2017*; *Li et al., 2019*).

It is estimated that the *Brassica-Arabidopsis* divergence occurred 14.5 to 20.4 million years ago (*Yang et al., 1999*). The *Brassica* genome experienced whole genome triplication (WGT) approximately 15.9 million years ago, followed by the divergence of *Brassica rapa* (AA, $2n = 20$) and *Brassica oleracea* (CC, $2n = 18$) approximately 4.6 million years ago (*Liu et al., 2014b*). *B. rapa* has three subgenomes with a different degree of gene loss due to whole genome triplication followed by the genome diploidization process, namely, least fractionated (LF), medium fractionated (MF1), and most fractionated (MF2) subgenomes (*Mun et al., 2009*). In the genus *Brassica*, *B. rapa* has the smallest genome, long seasonal varieties (e.g., turnip), and a similar degree of morphological diversity as *B. oleracea*. It is a model plant for use in studies on the genetics, genomics, and evolution of the *Brassica* species.

There is little known about the *OFP* gene family in *B. rapa*. The availability of information on the *B. rapa* genome allows us to analyze the *OFP* gene family on a genome-wide scale, which facilitates a better understanding on the roles of the *OFP* gene family in *B. rapa* growth and development. We conducted genome-wide analysis of the *OFP* gene family in *B. rapa*. We sought to determine: (1) the identification and characterization of *OFP* genes; (2) the duplication, cis-element distribution, and expression analysis of *OFP* genes; (3) the conserved motif and subcellular localization analysis of the OFP proteins.

## MATERIALS & METHODS

### Plant materials

*Brassica rapa* L. ssp. *pekinensis* var. BZ07-09 and *Brassica rapa* L. ssp. *pekinensis* var. BD05-272 are self-incompatible lines. *Brassica rapa* L. ssp. *pekinensis* var. weichunbai was synthesized by crossing 'BZ07-09' (maternal parent) and 'BD05-272' (paternal parent). The seeds of the three plants were obtained from the Vegetable Research Institute at the Weifang Academy of Agricultural Sciences, Shandong, China, and provided by Researcher T.H. Plant. Seeds were planted in the experimental fields of Weifang Academy of Agricultural Sciences, Weifang (36.62 degrees north latitude and 119.10 degrees east longitude), China, under natural conditions. Plant materials were sown in March (Spring), and leaf tissues were collected in April (Spring).

### Identification and renaming of *OFP* genes in *B. rapa*

*B. rapa* (v3.0) genome annotation information and protein sequences were downloaded from the BRAD database (http://brassicadb.cn/#/). A total of 19 OFP protein (AtOFP9 removed) sequences in *A. thaliana* were acquired from the TAIR database (https://www.arabidopsis.org/). Nineteen AtOFP protein sequences were used as query sequences for BLASTP against all *B. rapa* protein sequences with $E$-value $\leq$ 1e−10. Then the Hidden Markov Model (HMM) profile of the OVATE domain (Pfam: 04844) was used to search all *B. rapa* protein sequences using HMMER3 software. All OFP protein candidates from these searches were combined to make a non-redundant protein list. Finally, the non-redundant OFP protein sequences were analyzed domain using the NCBI Conserved Domain Search (https://www.ncbi.nlm.nih.gov/Structure/cdd/wrpsb.cgi?), HMMER Search (http://plants.ensembl.org/hmmer/index.html), and InterProScan (http://www.ebi.ac.uk/interpro/) to ensure the presence of the entire OVATE. All *OFP* genes in *B. rapa* were renamed using the standardized gene nomenclature for the *Brassica* genus (*Ostergaard & King, 2008*).

### Chromosome location, syntenic analysis, and duplication pattern identification

The genome file named 'Brapa_genome_v3.0_genes.gff3' contained the gene positions and structural information (http://www.brassicadb.cn/#/Download/) for all of the genes. The position and structure information of *OFP* genes was extracted from this file. The *OFP* gene chromosomal location was drafted by the MapInspect software based on their position information. SynOrths (http://www.brassicadb.cn/#/syntenic-gene/) was used to

determine the syntenic relationship between *B. rapa* and *A. thaliana* (*Cheng et al., 2012*), the relationship pairs were displayed by Circos (*Krzywinski et al., 2009*). The duplication types of *OFP* genes were confirmed using MCScanX, which could detect five duplication types, namely, singleton, proximal, dispersed, tandem, and WGD/segmental duplication (*Wang et al., 2012*).

## Phylogenetic tree construction, gene structure, and protein property analysis

The Phylogenetic tree was constructed using MEGA7 in line with the amino acid sequence similarity of BraOFP proteins with the neighbor-joining (NJ) method and bootstrap replication of 1,000 (*Kumar, Stecher & Tamura, 2016*). In addition, the phylogenetic relationships of BraOFP proteins along with AtOFP proteins were also produced using MEGA7. The exon–intron organization of the *OFP* genes was visualized using TBtools according to their structure information from the GFF file. The physicochemical properties of the OFP proteins, including isoelectric points (pI), molecular weight (MW) and grand average of hydropathy (GRAVY), were checked using ExPASy (https://web.expasy.org/protparam/).

## Subcellular localization, conserved motif analysis, and calculation of dN/dS values

The subcellular localization of the BraOFP proteins was predicted by Cell-PLoc 2.0 (http://www.csbio.sjtu.edu.cn/bioinf/Cell-PLoc-2/). The conserved motifs in BraOFP proteins were determined using MEME (https://meme-suite.org/meme/). The dN/dS values between the orthologous genes were analyzed to confirm the mode of selection. The protein sequences of each OFP orthologous pair between *B. rapa* and *A. thaliana* were aligned using MAFFT. The multiple sequence alignment results and the corresponding DNA sequences were then imported into PAL2NAL to convert into the relevant codon alignment. Finally, the codon alignment result was subjected to the calculation of non-synonymous (dN) and synonymous (dS) substitution rates using the codeml program in PAML (*Goldman & Yang, 1994*).

## Cis-element search in the *OFP* gene promoter regions

The promoter sequences of all *OFP* genes (the 2,000 bp upstream genomic sequences relative to the translation start codon) were retrieved from the *B. rapa* genome sequences. The cis-acting regulatory elements were scanned in the promoter sequences using PlantCARE (http://bioinformatics.psb.ugent.be/webtools/plantcare/html/).

## *OFP* gene expression analysis

Leaf tissues were collected from *B. rapa* L. ssp. *pekinensis* var. weichunbai and its parents (three samples). Each of the three samples had three biological replicates, and each biological replicate was composed of a combination of leaves from the three plants. Leaves were immediately frozen in liquid nitrogen for later high-throughput sequencing. Total RNA was extracted from the frozen leaves using the Trizol reagent kit (Invitrogen, Carlsbad, CA, USA), according to the manufacturer's instructions. RNA quality was assessed by

RNase free agarose gel electrophoresis, Agilent 2100 Bioanalyzer, and NanoPhotometer spectrophotometer. Nine cDNA libraries were constructed, and the library quality was checked by the NanoPhotometer spectrophotometer (*Nechifor-Boila et al., 2020*), Qubit2.0 Fluorometer (*Nechifor-Boila et al., 2020*) and Agilent 2100 bioanalyzer (*Panaro et al., 2000*). The cDNA libraries were sequenced on an Illumina sequencing platform (Illumina HiSeq^TM4000) by Gene Denovo Biotechnology Co. (Guangzhou, China). After removing low quality reads, the clean reads of three samples were mapped to *B. rapa* v3.0 reference genome using HISAT2 (*Kim, Langmead & Salzberg, 2015*). The clean reads mapped to the reference genome were assembled into the transcripts using String Tie software (*Pertea et al., 2015*). The fragment per kilobase of transcript per million mapped reads (FPKM) value was calculated to quantify the transcript expression abundance by RSEM (*Li & Dewey, 2011a*). To construct the heatmap using TBtools, the FPKM values of all *OFP* genes were converted to Z-values using the following equation: $\text{Z-value} = \frac{\log 2(\text{FPKM}) - \text{Mean}(\log 2(\text{FPKM}) \text{ of all samples})}{\text{standard deviation } (\log 2(\text{FPKM}) \text{ of all samples})}$.

## RESULTS

### Identification and chromosomal distribution of *BraOFP*s

A total of 29 *OFP* genes were identified in *B. rapa*, and each of them was renamed according to the standardized gene nomenclature. The details of these *BraOFP* genes, including the locus ID, genome position, coding sequence (CDS), and open reading frame (ORF) length are summarized in Table 1. The CDS length of the *BraOFP* genes varied from 266 bp to 1,210 bp, with an average of 778.52 bp. The ORF length of the *BraOFP* genes ranged from 266 bp to 2,259 bp, with an average of 860.52 bp. The majority of the *BraOFP* genes had the same CDS length as ORF, indicating that these genes had no intron. Only three *BraOFP* genes (BraA03.OFP5.b, BraA06.OFP7.a, and BraA10.OFP12.b) varied in CDS length and ORF length. The *BraOFP* genes were mapped to chromosomes A01-A10, and they were found to be unevenly distributed among the 10 chromosomes. Chromosomes 2 and 10 contained the largest number of *BraOFP* genes, comprising five members, followed by chromosome 5, with four members. Chromosomes 6 and 8 contained the fewest *BraOFP* genes, each with one member. Chromosomes 3, 4, and 9 each had three members, and chromosomes 1 and 7 each had two members (Fig. 1).

### Synteny and duplication analysis of *BraOFP*s

Syntenic gene pairs are orthologous gene pairs located in syntenic fragments between different species that originate from a common ancestor. Syntenic genes usually have similar functions, so syntenic analysis is used to explore the gene function of newly sequenced genomes or to reveal species genomic evolution. The syntenic relationships between the *OFP* genes of *B. rapa* and *A. thaliana* were confirmed using SynOrths of the BRAD database. *AtOFP2* and *AtOFP5* had three syntenic orthologs of *B. rapa*, which confirmed the occurrence of *Brassica* genome triplication. The other *AtOFP* genes had 0, 1, or 2 syntenic *BraOFP* orthologs, suggesting that *BraOFP* experienced gene loss. For example, *AtOFP6* had no syntenic *BraOFP* ortholog. Twenty-six out of 29 *BraOFP* genes had their syntenic orthologs in *A. thaliana*. Among the 26 *BraOFP* genes, 13 (50.0%) were

**Table 1 The information of *OFP* genes identified in *B. rapa*.** Chr means chromosome. CDS means coding sequence. ORF means open reading frame.

| Name | Locus ID | Chr | Gene start | Gene end | CDS length | ORF length |
|---|---|---|---|---|---|---|
| BraA02.OFP1.a | BraA02g000100.3C | A02 | 79062 | 79853 | 791 | 791 |
| BraA03.OFP2.a | BraA03g016190.3C | A03 | 7478278 | 7479204 | 926 | 926 |
| BraA04.OFP2.b | BraA04g021630.3C | A04 | 16198895 | 16199869 | 974 | 974 |
| BraA05.OFP2.c | BraA05g013910.3C | A05 | 7828677 | 7829627 | 950 | 950 |
| BraA10.OFP3.a | BraA10g016250.3C | A10 | 12585145 | 12586173 | 1028 | 1028 |
| BraA03.OFP4.a | BraA03g028060.3C | A03 | 14017951 | 14018292 | 341 | 341 |
| BraA09.OFP4.b | BraA09g063630.3C | A09 | 43865357 | 43866262 | 905 | 905 |
| BraA01.OFP5.a | BraA01g010340.3C | A01 | 5317122 | 5318105 | 983 | 983 |
| BraA03.OFP5.b | BraA03g048500.3C | A03 | 24609733 | 24610785 | 904 | 1052 |
| BraA08.OFP5.c | BraA08g013140.3C | A08 | 11325678 | 11326643 | 965 | 965 |
| BraA06.OFP7.a | BraA06g029810.3C | A06 | 20612591 | 20614850 | 1210 | 2259 |
| BraA07.OFP7.b | BraA07g002210.3C | A07 | 1673203 | 1674150 | 947 | 947 |
| BraA02.OFP8.a | BraA02g008340.3C | A02 | 3967722 | 3968342 | 620 | 620 |
| BraA10.OFP8.b | BraA10g020720.3C | A10 | 14938955 | 14939596 | 641 | 641 |
| BraA02.OFP10.a | BraA02g009640.3C | A02 | 4598799 | 4599287 | 488 | 488 |
| BraA10.OFP10.b | BraA10g019070.3C | A10 | 14087972 | 14088238 | 266 | 266 |
| BraA01.OFP11.a | BraA01g021760.3C | A01 | 11947484 | 11948128 | 644 | 644 |
| BraA09.OFP12.a | BraA09g064430.3C | A09 | 44194487 | 44195170 | 683 | 683 |
| BraA10.OFP12.b | BraA10g004230.3C | A10 | 2231910 | 2233990 | 899 | 2080 |
| BraA02.OFP13.a | BraA02g001490.3C | A02 | 774145 | 774816 | 671 | 671 |
| BraA10.OFP13.b | BraA10g031620.3C | A10 | 19700352 | 19701140 | 788 | 788 |
| BraA02.OFP14.a | BraA02g026230.3C | A02 | 15718578 | 15719465 | 887 | 887 |
| BraA07.OFP14.b | BraA07g043220.3C | A07 | 28719909 | 28720769 | 860 | 860 |
| BraA04.OFP15.a | BraA04g025670.3C | A04 | 18379375 | 18380160 | 785 | 785 |
| BraA05.OFP15.b | BraA05g009320.3C | A05 | 4958275 | 4959087 | 812 | 812 |
| BraA05.OFP16.a | BraA05g012270.3C | A05 | 6656598 | 6657311 | 713 | 713 |
| BraA09.OFP18.a | BraA09g043680.3C | A09 | 33748540 | 33749316 | 776 | 776 |
| BraA04.OFP19.a | BraA04g025660.3C | A04 | 18367416 | 18367976 | 560 | 560 |
| BraA05.OFP19.b | BraA05g009340.3C | A05 | 4973405 | 4973965 | 560 | 560 |

assigned to the LF subgenome, five (19.2%) to the MF1subgenoeme, and eight (30.8%) to the MF2 subgenoeme. The syntenic relationship pairs of between the *OFP* genes from *B. rapa* and *A. thaliana* are shown were displayed in Fig. 2. In addition, the three paralogous copies of *BraOFP2* and *BraOFP5* were simultaneously retained on the LF, MF1 and MF2 subgenomes. For *BraOFP1* and *BraOFP6*, none of the three paralogous copies were retained. For the other *BraOFPs*, one or two of the three paralogous copies were retained.

According to the results of syntenic analysis and reciprocal BLASTP analysis, we established 29 orthologous gene pairs between *OFP* genes from *B. rapa* and *A. thaliana*. Synonymous (dS) and nonsynonymous (dN) values were calculated to explore the selective pressure on the *BraOFP* genes. Generally, a dN/dS value greater than 1 indicates positive selection, a value less than 1 indicates a purifying selection, and a value equal to 1 indicates neutral selection (*Nekrutenko, Makova & Li, 2002*). All orthologous gene pairs had a dN/dS

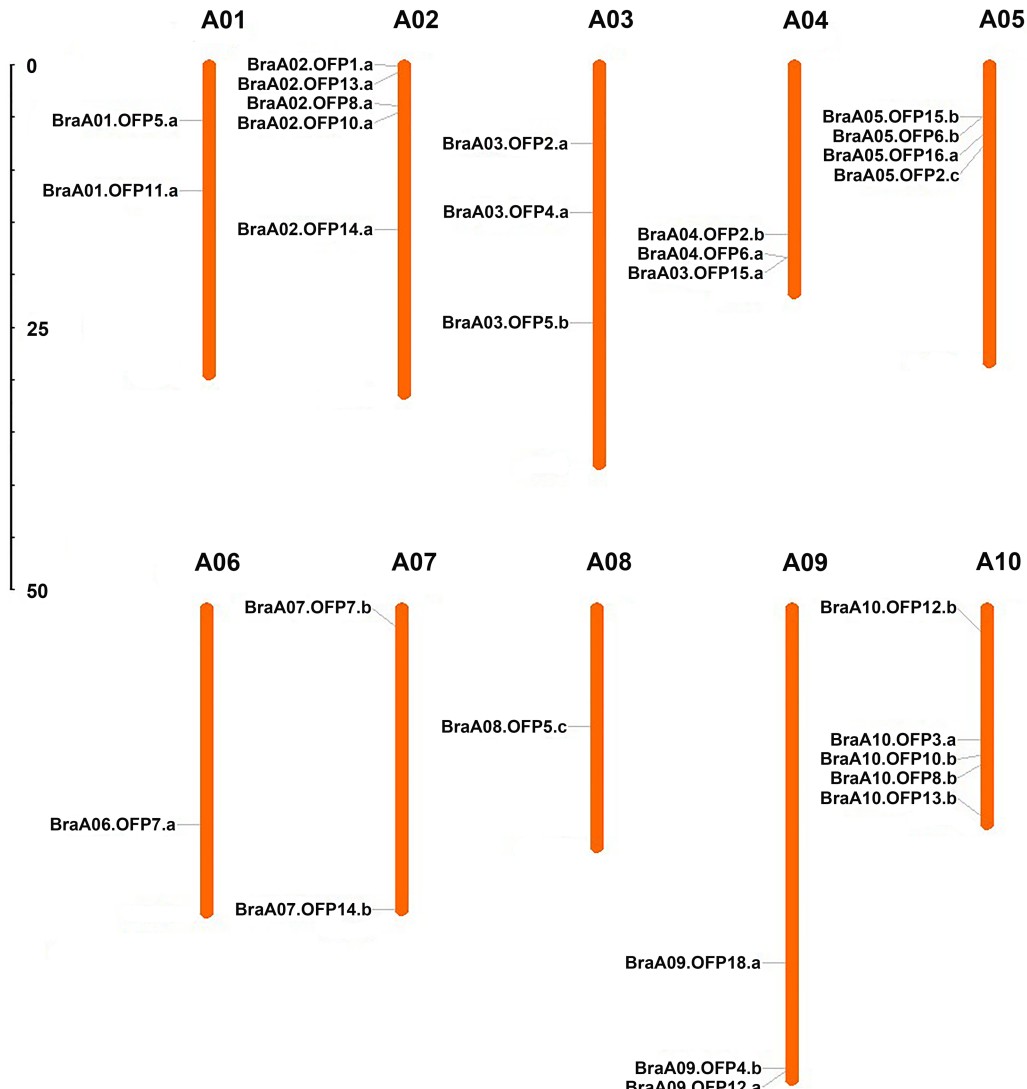

**Figure 1** **The chromosomal location of *OFP* genes in *B. rapa*.** The scale is in megabases (Mb).

ratio of less than 0.5, showing that a strong purifying selection acted on these *BraOFP* genes (Table S1). The lowest dN/dS ratio is only 0.0108, suggesting that BraA03.OFP4.a experienced an extremely strong purifying selection. These findings indicated that *BraOFP* genes may preferentially conserve structure and function under selective pressure.

The duplication types of *BraOFP* genes were identified and classified using MCScanX, and two duplicated types of *BraOFP* genes were found, namely dispersed and WGD/segmental duplication genes (Table S2). We found that 82.8% of *BraOFP* genes were generated by WGD/segmental duplication and only 17.2% of *BraOFP* genes were produced by dispersed duplication, suggesting that the duplication pattern of *BraOFP* genes was high WGD and low dispersed duplication. Our results were consistent with those of previous studies, indicating that the main cause of *BraOFP* gene expansion was also WGD (*Liu et al.,*

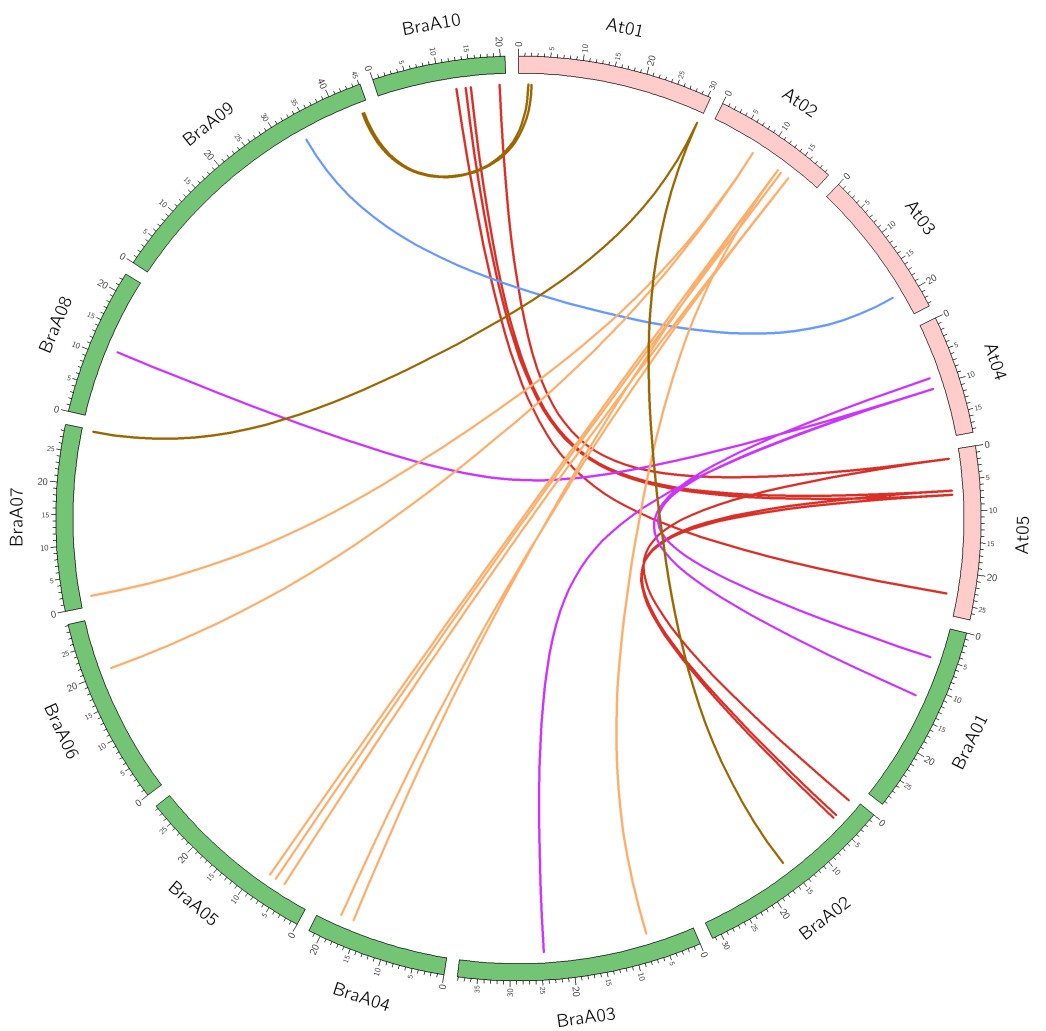

**Figure 2** **The synteny analysis of *OFP* genes between *B. rapa* and *A. thaliana*.** Numbers on the chromosomes represent mega base pairs. The positions of *OFP* genes on the chromosomes are indicated.

*2014a*). The ancestral *Brassica* genome experienced a triplication event, which contributed to the expansion of the *OFP* gene family in *B. rapa*.

## Phylogenetic and gene structural analysis of *BraOFPs*

The phylogenetic tree of 29 BraOFP proteins was constructed, and were grouped into 3 distinct subfamilies (subfamilies I, II, III) (Fig. 3A). Subfamily-III, with 11 members, was the largest, followed by subfamily-I with ten members, and subfamily-II with eight members. A sister pair indicated the closest genetic relationship in the phylogenetic tree, and 11 sister pairs were found, including three, four, and four pairs in subfamilies I, II, and III, respectively. Almost all sister pairs were duplication gene pairs (paralogous gene pairs), with the exception of BraA02.OFP1.a and BraA10.OFP3.a. BraA02.OFP1.a and BraA10.OFP3.a were clustered together to become a sister pair, suggesting a close evolutionary relationship between BraA02.OFP1.a and BraA10.OFP3.a.

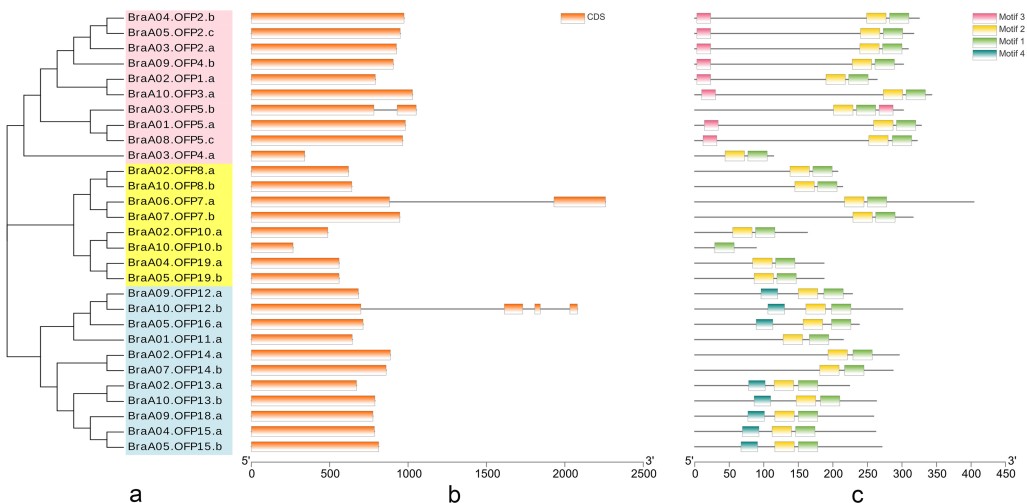

**Figure 3** **The phylogenetic relationship (A) and the motif composition (C) of OFP proteins, and the exon–intron organization (B) of *OFP* genes in *B. rapa*.** (A) Twenty-nine BraOFP proteins are grouped into three distinct subfamilies. (B) Exons and introns are indicated by orange boxes and single gray lines, respectively. (C) Four conserved motifs are shown in different colors.

We conducted exon–intron organization analysis to obtain insights into the structural variation of the *BraOFP* genes. Our results showed that only three *BraOFP* members had introns, which belonged to subfamilies I, II, and III, respectively (Fig. 3B). *AtOFP5*, *AtOFP7* and *AtOFP12* had no intron, while their *B. rapa* orthologous genes (BraA03.OFP5.b, BraA06.OFP7.a, BraA10.OFP12.b) had introns. *Brassica* and Arabidopsis originated from a common ancestor, suggesting that the *BraOFP* genes experienced an intron gain event. Intron gain is an important evolutionary mechanism that brings about gene structural complexity and diversity, and promotes genetic functional divergence and diversity during the evolution of multi-gene families (*Li et al., 2009*). *BraOFP5* had three paralogs (BraA01.OFP5.a, BraA03.OFP5.b, and BraA08.OFP5.c), only BraA03.OFP5.b had a intron, and intron gain made the gene structure of BraA03.OFP5.b different form the other two paralogs, which might mean the functional segregation of the *BraOFP5* gene. As paralogous gene pairs, BraA06.OFP7.a had a intron when compared with BraA07.OFP7.b. The gene structure of BraA10.OFP12.b was more complex, and it had three introns compared with its paralogous BraA09.OFP12.a. Thus, it is likely that intron gain events may have occurred during the structural evolution of *BraOFP* paralogues.

## Physicochemical property, subcellular localization and conserved motifs of BraOFPs

The average length of the BraOFP proteins was 259 aa, the shortest was BraA10.OFP10.b with 88 aa, and the longest was BraA06.OFP7.a with 403 aa. The molecular weight of BraOFP proteins varied from 10.31 KD to 44.83 KD, with an average of 29.40 KD. Sixty-two percent of BraOFP proteins had relatively high isoelectric point (pI) (pI > 7), and all the BnaGH3 proteins had an instability index exceeding 40, indicating that these proteins could be unstable. All the BraOFP proteins exhibited GRAVY values in the negative range,

revealing their hydrophilic natures. Among these, BraA01.OFP5.a, BraA08.OFP5.c, and BraA02.OFP14.a were thought to be the most hydrophilic with GRAVY values less than −1 (Table S3).

Transcription regulators are generally localized in the nucleus to control the expression of target genes. Most AtOFP members were thought to contain a nuclear localization signal (*Hackbusch et al., 2005*). The subcellular location of the protein is important for predicting their potential functions and possible biological significance. The subcellular localization of the BraOFP members was predicted using Cell-PLoc 2.0. Our results revealed that all members were predicted to be in the nucleus (Table S3). In addition, BraA06.OFP7.a and BraA02.OFP10.a were also found in other locations (chloroplast and cell membrane), indicating that these two genes had other functions. The presence of all BraOFPs in the nucleus may support their roles as transcription factors, and two members were also located in other subcellular compartments, suggesting dynamic properties that are undiscovered or unknown.

All BraOFP protein sequences were input into MEME for motif analysis to better understand the motif composition. Four motifs were identified and designated as motifs 1 to 4 (Fig. 3C). The logos of the four motifs are shown in Fig. 4. Motif 1 was found in all BraOFPs, indicating that it was indispensable for BraOFP function. Except BraA10.OFP10.b with shortest sequence, the remaining 28 proteins all contained motif 2. BraA10.OFP10.b had only one motif and it was the simplest OFP protein in *B. rapa*. The members clustered together in the phylogenetic tree had a similar conserved motif distribution. Motif 3 was specific to subfamily I, whereas motif 4 was specific to subfamily III, suggesting the functional similarities within the subfamilies. The characteristics of different motif compositions within BraOFPs could reflect functional inconsistencies.

## Phylogenetic analysis of OFP proteins in *B. rapa* and *A. thaliana*

To better assess OFP phylogenetic relationships, we constructed a phylogenetic tree of 48 OFPs from *B. rapa* and *A. thaliana* and they were divided into six distinct subgroups (I, II, III, IV, V and VI) (Fig. 5). Subgroup V constituted the largest clade containing 18 members with 11 BraOFPs and 7AtOFPs. Subgroup I formed the second largest clade, including15 members with 10 BraOFPs and 5 AtOFPs. Subgroup VI comprised the smallest OFP class, and only contained two AtOFPs. In addition, subgroup II contained two BraOFPs and two AtOFPs. Subgroup III contained six members with four BraOFPs and two AtOFPs. Subgroup IV contained two BraOFPs and one AtOFPs. A total of 17 sister pairs were found, including 11 orthologous gene pairs between *B. rapa* and *A. thaliana*, five paralogous BraOFP pairs, and one pair of AtOFPs. AtOFP17 and AtOFP20 were clustered together to form a sister pair, forming a distinct subgroup. It was reported that AtOFP20 was most closely related to AtOFP17, and they were putative paralogues occurring within segmental duplication blocks (*Liu et al., 2014a*). A few AtOFP/BraOFP proteins showed a 1:2 orthologous relationship, such as AtOFP7/BraA06.OFP7.a and BraA07.OFP7.b, and such an orthologous relationship demonstrated the occurrence of *Brassica* genome triplication. A few AtOFPs had only one orthologous BraOFP, indicating well conserved

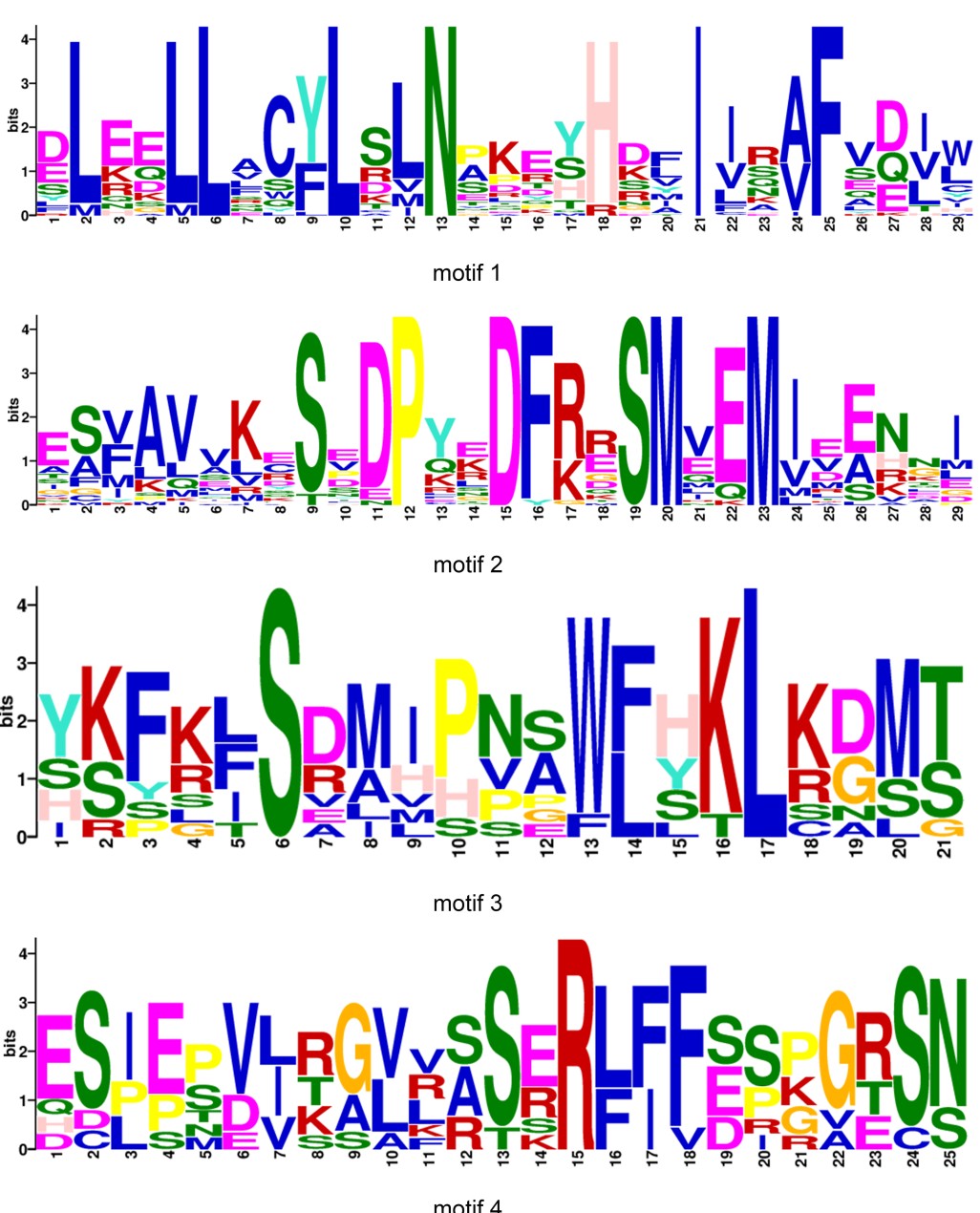

**Figure 4** **The logos of four motifs.** The *y*-axis represents information contents in bits, and the *x*-axis represents the motif width.

functions. The biological functions of AtOFPs are known, which may be used to explore the functions of their corresponding orthologous BraOFP.

The multiple sequence alignment of 48 OFP proteins was achieved by MAFFT, and the conserved domain logo was shown using WebLogo. The results indicated that the proteins all had an entire OVATE domain located in the C-terminus (Fig. 6). The domain contained some conserved sites, including P, D, F N, H, and I, and they were relatively conserved

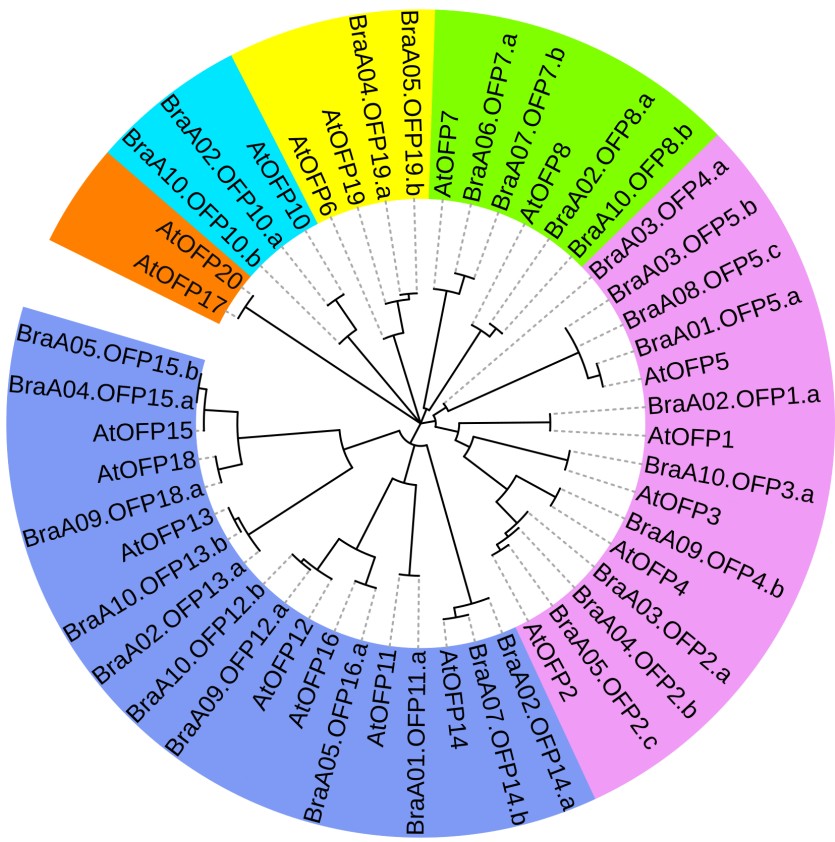

**Figure 5  The phylogenetic relationships of 48 OFP proteins from *B. rapa* and *A. thaliana*.** The phylogenetic tree is divided into six distinct subgroups in different colors.

sites. The domain also had a highly conserved site, L, which may be necessary for OFP protein function.

## The cis-acting elements in the promoters of *BraOFP*s

Cis-acting elements within the 5ʹregulatory region of the gene sequences are vital molecular switches involved in regulating the levels of both temporal and spatial expression of the functional genes, and controlling various biological processes (*Himani et al., 2014*). The identification of cis-acting elements can improve our understanding of *BraOFP* expression and regulation. PlantCARE was used to search for the cis-acting elements within the promoter regions of the *BraOFP* genes, and the results are shown in Fig. 7. We found that all *BraOFP* genes had the light responsive-related cis-acting elements, indicating that *BraOFP* genes could be photo-responsive and light regulatory trans-acting factors bound to these light responsive-related cis-acting elements, modulating *BraOFP* gene activity in response to light. Light, as a major source of energy and an environmental signal, is crucial for plant growth and development from seed germination to maturation, and *BraOFP* genes maybe play an important role in this process.

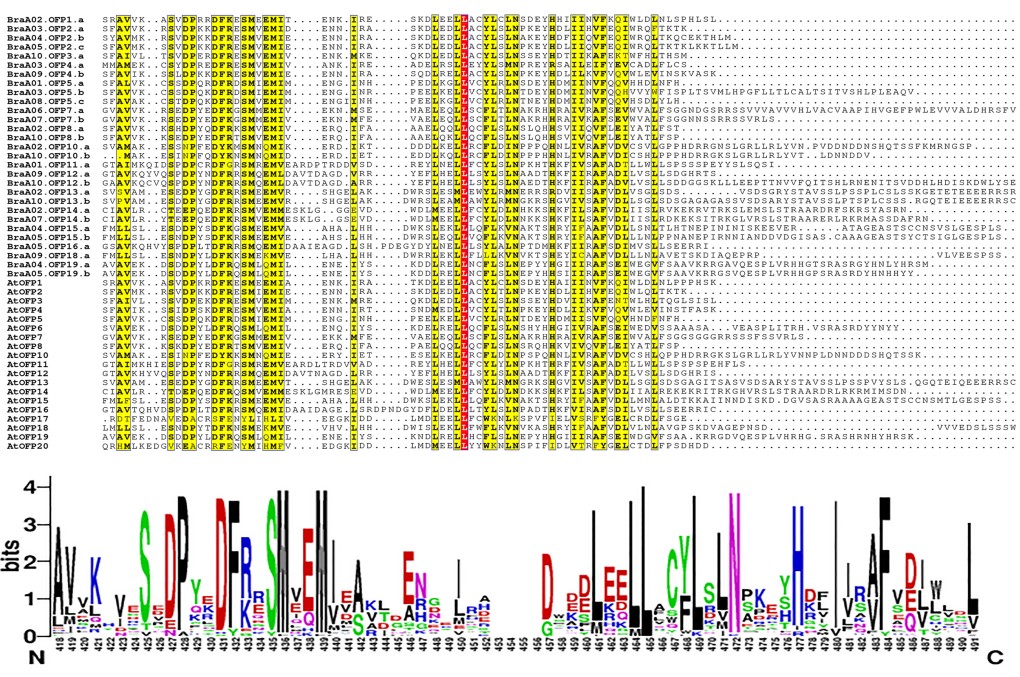

**Figure 6** **The multiple sequence alignment of 48 OFP proteins from *B. rapa* and *A. thaliana*.** The yellow regions show the relatively conserved sites, and a red region shows a highly conserved site. The motif logo shows an entire OVATE domain, the *y*-axis represents information contents in bits, and the *x*-axis represents the motif width.

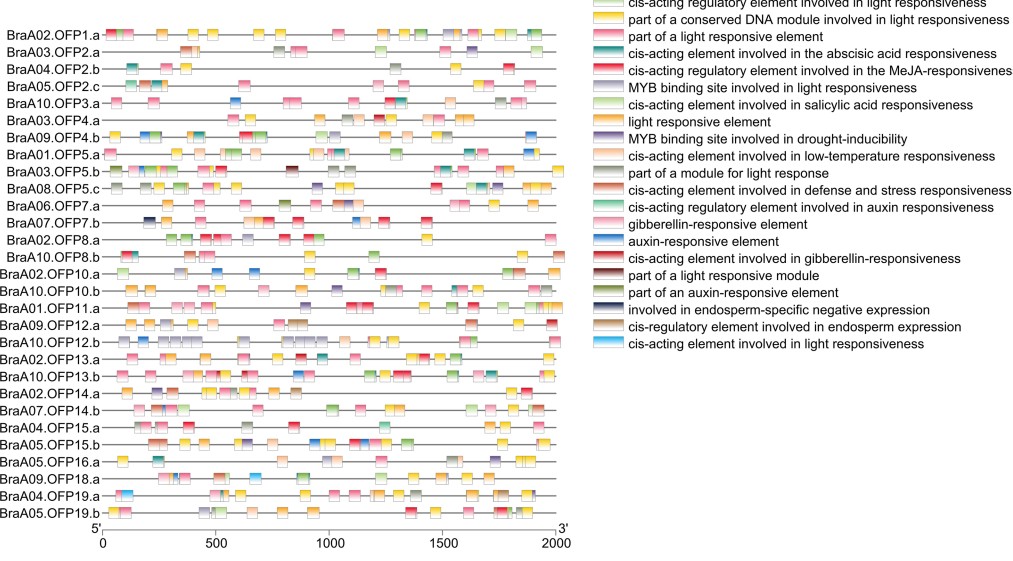

**Figure 7** **The cis-acting element distribution within the promoters of the *BraOFP* genes.** Twenty-one cis-acting elements are shown in different colors.

All *BraOFP* genes also had phytohormone-related elements, showing that *BraOFP* genes were involved in phytohormone signaling pathways and their expression was controlled by the phytohormone. Phytohormone-related cis-acting elements contained salicylic acid-responsive element (10 *BraOFP*s), gibberellin-responsive element (14 *BraOFP*s), auxin-responsive element (16 *BraOFP*s), MeJA-responsive element (18 *BraOFP*s), and abscisic acid-responsive element (20 *BraOFP*s) (Fig. 8). BraA05.OFP2.c, BraA10.OFP3.a, BraA03.OFP5.b, and BraA10.OFP13.b each had four kinds of hormone response elements, while BraA03.OFP4.a, BraA03.OFP2.a, BraA10.OFP12.b, and BraA02.OFP14.a each had one kind of hormone response element. These suggested that *BraOFP* genes could take part in diverse aspects of plant growth and development. Additionally, stress-related cis-acting elements were discovered in the promoters of *BraOFP* genes. Eleven, fourteen, and fifteen *BraOFP* genes had drought-related elements, defense responsive elements and low-temperature responsive elements, respectively, indicating that these *BraOFP* genes maybe enhance the stress resistance of *B. rapa*. Finally, BraA07.OFP7.b, BraA09.OFP12.a, BraA02.OFP14.a, and BraA04.OFP19.a had cis-acting elements involved in endosperm expression, which suggested they might be involved in the seed development of *B. rapa*.

### The gene expression analysis of *BraOFP*s

The heterosis phenomenon is common in higher plants and the hybrid generation is superior to its parents in yield and stress resistance. To explore the effect of hybridization on the expression of *OFP* genes, we compared the expression levels of *OFP* genes between the hybrid and its parents (Table S4). Relative to its parents, the expression levels of six *OFP* genes (BraA02.OFP13.a, BraA10.OFP13.b, BraA02.OFP14.a, BraA04.OFP15.a, BraA05.OFP16.a, and BraA04.OFP19.a) were significantly increased in the hybrid (Fig. 9), suggesting that hybridization could gave rise to changes in *OFP* gene expression, which may contribute to the formation of heterosis. Previous studies showed that differential gene expression was closely related to the formation of heterosis in the Chinese cabbage hybrid (*Wu, Cao & Dong, 2004*; *Lin et al., 2012*).

We compared the expression patterns of the paralogous genes among three samples to investigate whether the duplicated genes had different expression patterns. There were eleven paralogous gene sets in total, and one paralogous gene set (BraA09.OFP12.a and BraA10.OFP12.b) had no expression in all three samples. Only two paralogous gene sets (BraA02.OFP13.a and BraA10.OFP13.b, BraA04.OFP19.a and BraA05.OFP19.b) had the same expression patterns, and the remaining paralogous gene sets had different expression patterns. For example, BraA01.OFP5.a, BraA03.OFP5.b, and BraA08.OFP5.c had the highest expression levels in the paternal parent, hybrid and maternal parent, respectively. The paralogous gene sets showed discrepant expression patterns, which could be on account of the functional divergence of duplicated genes.

## DISCUSSION

OFP proteins function as transcriptional factors and are found exclusively in the plant kingdom. They are responsible for regulating multiple aspects of plant growth and development. The *OFP* gene family has been reported in various plant species, however,
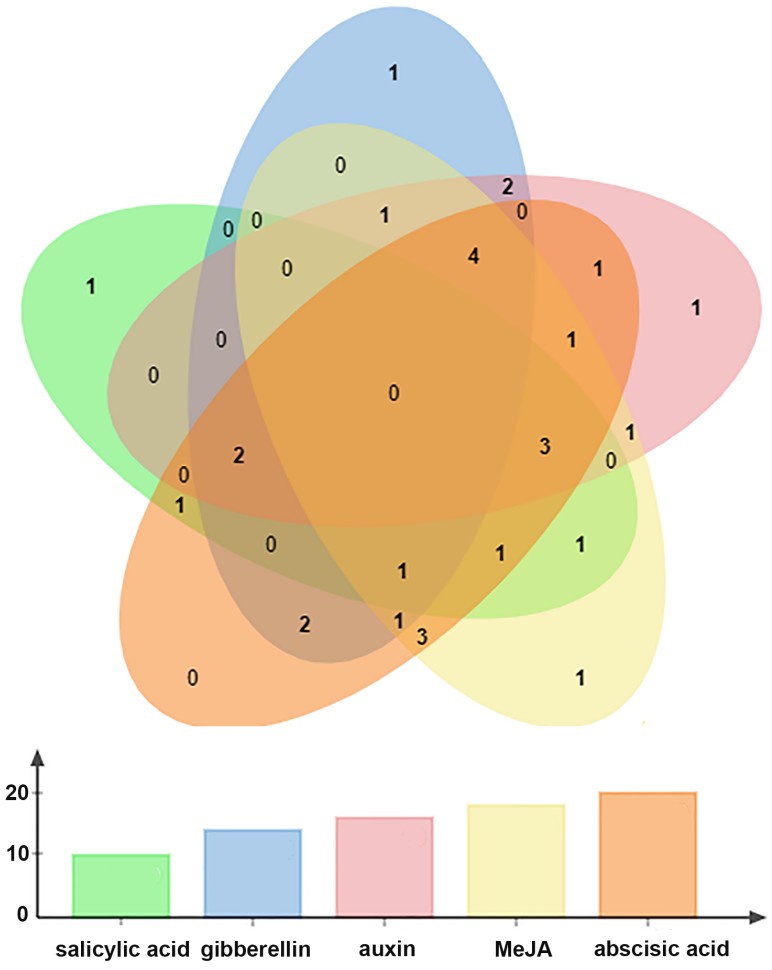

**Figure 8** **The Venn diagram showing the number of phytohormone-related elements in the promoters of the *BraOFP* genes.** Phytohormone-related elements are represented by different colors. The $x$-axis represents the types of element, and the $y$-axis represents the number of *BraOFP* genes.

little is known about this gene family in *B. rapa*. We provided substantial information on the *OFP* gene family in *B. rapa*, including member identification, chromosomal location, gene structural organization, gene duplication and syntenic analysis, and cis-element and gene expression analysis. Our results provide a framework for further study on the biological functions of *OFPs* in *Brassica* species.

## The member expansion and loss of the *OFP* gene family in *B. rapa*

The whole genome duplication (WGD) or polyploidization is a major driving force in the evolution of angiosperms, resulting in the expansion of genome content and the diversity of gene function (*Magadum et al., 2013*). *B. rapa* is a mesopolyploid crop that experienced a whole genome triplication event after its divergence from *Arabidopsis*, which was followed by gene loss, retention, and recombination. Genes that encode the proteins related to signal transduction or transcriptional regulation were largely retained (*Wang et al., 2011b*). Some

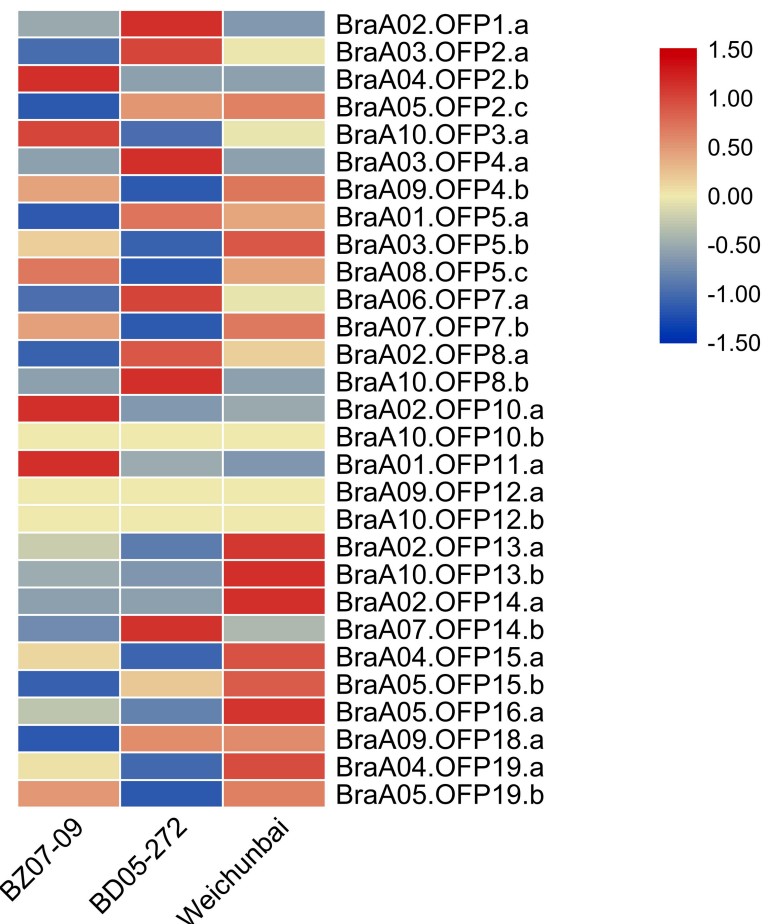

**Figure 9 Heatmap of expression level of 29 *BraOFP* genes in the hybrid and its parents.** Red and blue color scale indicates high and low expression levels, respectively. Yellow color indicates barely no gene expression.

gene families in *B. rapa* expanded mainly through segmental duplication/WGD to produce the large number of members, such as the Homeobox gene family, CCCH zinc finger family, or the CDPK-SnRK gene family (*Wu et al., 2017*; *Khan et al., 2018*; *Pi et al., 2018*). Our results revealed that the majority of *BraOFP* genes (83%) belonged to WGD/segmental type based on duplicated analysis, and only five *BraOFP* genes belonged to dispersed type (Table S2). These results suggest that WGD played a predominant role in the expansion of the *OFP* gene family in *B. rapa*. Previous studies showed that WGD also contributed the expansion of the *OFP* gene family in apple, grape, and rice (*Liu et al., 2014a*; *Wang et al., 2018*; *Li et al., 2019*).

Due to the whole genome triplication event, the number of *OFP* genes in *B. rapa* should be three times than that of *Arabidopsis*. In theory, 57 *OFP* genes exist in *B. rapa*. But, only 29 *OFP* genes were identified in *B. rapa* (Table 1), suggesting that there was extensive gene loss after genome triplication. Our syntenic analysis also confirmed this. For example, *AtOFP3*, *AtOFP11*, and *AtOFP16* each had only one syntenic orthologous *BraOFP*, and

even *AtOFP1* and *AtOFP6* had no syntenic orthologous *BraOFP* (Fig. 2). We found that 26 *BraOFP* genes showed syntenic relationships with 15 *AtOFP* genes, indicating that the number of duplicated *BraOFP* genes was only approximately twice that of the *AtOFP* genes. These results were consistent with an earlier study that showed that the triplicated *B. rapa* genome contained approximately twice the number of genes as *Arabidopsis* on account of genome shrinkage (*Mun et al., 2009*). Only *BraOFP2* and *BraOFP5* retained three copies, implying that these two genes might have irreplaceable functions. *OFP2* was shown to have vital effects on phytohormonal homeostasis and vasculature development in rice (*Schmitz et al., 2015*), and *OFP5* is required for the normal development of the female gametophyte in *Arabidopsis* (*Pagnussat, Yu & Sundaresana, 2007*). Although there was a shrinkage of some members, the whole genome triplication indeed expanded the member number of the *OFP* gene family, and the retained members could be the key regulators with comprehensive and vital functions during *B. rapa* growth and development.

*Wang et al. (2011a)* reported that functional redundancy existed among *AtOFP* genes, so the losses of *BraOFP* genes may have positive significance for *B. rapa* growth and development. Two main mechanisms caused genes to disappear from the genome (*Albalat & Cañestro, 2016*). First, the gene loss was the consequence of an abrupt mutational event, such as the mobilization of a transposable element. Second, the gene loss was the consequence of a slow process of mutation accumulation during the pseudogenization. Plants could better adapt to the environment through gene losses. For example, the loss of *SCR* and *SRK* genes laid the foundation for the evolution from obligate outcrossing on account of self-incompatibility to a self-fertilization system in *A. thaliana* (*Shimizu et al., 2008*). The deletion of a gene encoding an anthocyanin pathway enzyme was closely related to the transition from blue flowers to red flowers, which induced phenotypic differences among *Andean Iochroma* species (*Smith & Rausher, 2011*). The loss of *BraOFP* genes has possibly led *B. rapa* to better adapt to the environment.

### The emergence of introns in *BraOFP* genes

*Liu et al. (2014a)* found that almost all *OsOFP* genes were without introns, except *OsOFP10*, which had a small intron. Similarly, only six of thirty-five *OFP* genes had introns in radish, and all *OFP* genes in apple were intron-free (*Li et al., 2019*; *Wang et al., 2020*). We found that the overwhelming majority of *BraOFP* genes were without introns, while BraA03.OFP5.b, BraA06.OFP7.a, and BraA10.OFP12.b had introns (Fig. 3B). The appearance of intron created differences in the gene structure between paralogous *BraOFP* genes. Paralogous gene pairs, as the products of gene duplication, were inclined to diverge in regulatory and coding regions. Structural divergences in the genes were crucial to the evolution of paralogous gene pairs, which were produced by three types of mechanisms, namely, exonization/pseudoexonization, exon/intron gain/loss, and insertion/deletion (*Xu et al., 2012*). Intron gain occurred when a piece of unrelated, exotic nucleotide sequence was inserted into an exon and gave rise to exon fission. This led to the generation of proteins with distinct sequence features, followed by the acquisition of new functions. Therefore, BraA03.OFP5.b, BraA06.OFP7.a, and BraA10.OFP12.b may have the potential to generate proteins with new functions distinct from their paralogous

genes. All *AtOFP* genes were without introns, with the exception of *AtOFP17*. There were 29 orthologous *OFP* gene pairs between *B. rapa* and *A. thaliana*, and three pairs (10%, *AtOFP5/BraA03.OFP5.b*, *AtOFP7/BraA06.OFP7.a*, and *AtOFP12/BraA10.OFP12.b*) had differences in exon-intron organization. There were 15 paralogous *OFP* gene sets in *B. rapa*, and four sets (27%, *BraA03.OFP5.a /BraA03.OFP5.b*, *BraA03.OFP5.b /BraA03.OFP5.c*, *BraA06.OFP7.a/BraA06.OFP7.b*, and *BraA10.OFP12.a /BraA10.OFP12.b*) had differences in exon-intron organization. The divergences of exon-intron organization also occurred in orthologous gene pairs, but the rates were usually lower than paralogous gene pairs, which was consistent with results from a previous study (*Xu et al., 2012*).

Alternative splicing is a post-transcriptional regulatory process and can result in the generation of multiple transcripts from a single gene, influencing the location and function of the protein (*Reddy, 2007*). The presence of introns allows transcript diversity to be generated by alternative splicing. Alternative splicing may occur in *BraA03.OFP5.b*, *BraA06.OFP7.a*, and *BraA10.OFP12.b*. These three genes could produce transcript isoforms that were translated into new proteins, implying the functional divergence between paralogous gene pairs. Relative to *BraA03.OFP5.b* and *BraA06.OFP7.a*, *BraA10.OFP12.b* had three introns, so its alternative splicing forms were more diversified, producing more alternative splicing transcripts and greater functional diversification. *Wang et al. (2011a)* found that *AtOFP12* showed ubiquitous expression across various tissues, which suggested that it was important to plant growth and development. Gene expression is regulated at multiple levels, and alternative splicing can regulate gene expression at post-transcriptional levels by increasing or reducing gene expression levels. The expression levels of *BraA03.OFP5.b*, *BraA06.OFP7.a*, and *BraA10.OFP12.b* may be controlled by alternative splicing, indicating the diversity of regulatory mechanisms.

Intron gain could be an important evolutionary mechanism to produce genetic structural variety and complexity, and facilitate functional divergence and diversity during the evolutionary process of the *OFP* gene family in *B. rapa*.

## The expression changes of *BraOFP* genes may be beneficial to the formation of heterosis

Heterosis can raise crop yield, enhance disease and stress resistance, and improve crop quality. More research has shown that the differential gene expression promoted the formation of heterosis. *Zhai et al. (2013)* found that several differentially expressed genes between the rice hybrid and its parents were mapped to quantitative trait loci for yield and root traits and were involved in plant hormone signal transduction and carbohydrate metabolism pathways. These may significantly contribute to yield heterosis. The differentially expressed genes in wheat hybrids were involved in important biological pathways such as photosynthesis and carbon fixation, which promoted the formation of heterosis (*Liu et al., 2018*). The differentially expressed genes in sorghum hybrids were closely related to the formation of grain yield advantage (*Jaikishan et al., 2019*).

*B. rapa* L. ssp. *pekinensis* var. weichunbai had obvious heterosis relative to its parents. It has higher yield, stronger disease resistance, and richer nutrients (*Han et al., 2018*). The expression levels of six *OFP* genes in the hybrid were significantly improved relative to

the parents and may play a role in the formation of heterosis (Table S4). *OFP13*, *OFP15*, and *OFP16* functioned in the ERECTA (ER) signaling pathway to control plant growth and development, including stomata initiation, silique and inflorescence morphology, and response to environmental stresses (*Wang et al., 2019*). *OFP19* played a critical role in modulating brassinosteroid signaling and determining the cell division pattern. The overexpression of *OFP19* caused the production of thicker leaves, which may positively affect the yield of Chinese cabbage (*Yang et al., 2018*).

The expression levels of certain genes in the hybrids were significantly altered and there were even some specifically expressed genes, which may be due to the changes in gene expression regulatory networks in the heterozygous state, which stimulated new regulatory mechanisms. These mechanisms altered the levels of gene expression, followed by changes in a variety of physiological and biochemical reactions, which led to trait changes and the formation of advantageous phenotypes.

## CONCLUSIONS

In this study, we performed genomic identification and analysis of the *OFP* gene family in *B. rapa*, with 29 confirmed *OFP* genes. These *OFP* genes were unevenly distributed among 10 chromosomes and only three *BraOFP* genes had introns. Syntenic analysis showed that *BraOFP* went through gene loss after *Brassica* genome triplication. Additionally, WGD greatly contributed the expansion of the *OFP* gene family in *B. rapa*, which was consistent with previous studies. All *BraOFP* genes were found to have light responsive-related and phytohormone-related cis-acting elements. Finally, expression analysis revealed that the expression levels of six *OFP* genes had an obvious increase after hybridization, and the paralogous gene sets showed discrepant expression patterns among the hybrid and its parents. Our study provides valuable information on the *OFP* gene family in *B. rapa* to provide a foundation to further investigate the roles of *OFP* genes in the formation of heterosis.

### Funding
This work was supported by the Doctoral Research Foundation of Weifang University (2020BS24). The funders had no role in study design, data collection and analysis, decision to publish, or preparation of the manuscript.

### Grant Disclosures
The following grant information was disclosed by the authors:
Doctoral Research Foundation of Weifang University: 2020BS24.

### Competing Interests
The authors declare there are no competing interests.

## Author Contributions

- Ruihua Wang conceived and designed the experiments, performed the experiments, analyzed the data, prepared figures and/or tables, authored or reviewed drafts of the paper, and approved the final draft.
- Taili Han performed the experiments, authored or reviewed drafts of the paper, provided and managed the plant materials, and approved the final draft.
- Jifeng Sun and Ligong Xu performed the experiments, authored or reviewed drafts of the paper, planted and collected the plant materials, and approved the final draft.
- Jingjing Fan and Hui Cao performed the experiments, authored or reviewed drafts of the paper, and approved the final draft.
- Chunxiang Liu conceived and designed the experiments, performed the experiments, prepared figures and/or tables, authored or reviewed drafts of the paper, and approved the final draft.

## Data Availability

Raw RNA-seq data is available at the Sequence Read Archive (SRA): PRJNA660555. The sequences are also available in the Supplemental Files.

## Supplemental Information

Supplemental information for this article can be found online at http://dx.doi.org/10.7717/peerj.10934#supplemental-information.

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
