# Peer review of "Genome-wide identification and characterization of the OFP gene family in Chinese cabbage (Brassica rapa L. ssp. pekinensis)"

_PeerJ, doi:10.7717/peerj.10934_

## Round 0.1 · original submission · Major Revisions

Your paper needs major revision. Revise your paper according to reviewers' reports. Improve the quality of your paper.

Reviewer 1 ·

Basic reporting

The manuscript is not well written. The authors organized the paper so wordy and unclear. Extensive editing of English language is required. Besides, the introduction failed to provide sufficient background and include all relevant references.

Experimental design

No comment.

Validity of the findings

The conclusions are not supported by the results. The authors found that the expression levels of six OFP genes are higher in the Brassica rapa hybrid than in the parents. So, they summarized that the altered expression contributes to the heterosis. It's so weak. The changed expression of the genes in question may also contribute to hybrid weakness.

Additional comments

In this manuscript, the authors described the chromosal distribution of OFPs and their gene structure. They found all BraOFP genes owned light responsive-related and phytohormone-related cis-acting elements. Moreover, six OFP genes expressed at higher level in the hybrid than in its parents. It is likely that the findings in this paper may attract the interest of some readers in this field. However, I have some suggestions.
Major points:
1. The data don't support their conclusion that the altered expression contributes to the heterosis.
2. The manuscript is unreadable. Extensive editing of English language is required to make the manuscript clear.
3. The introduction failed to provide sufficient background.
Minor points:
1. Re-write the title.
2. Line 19: Define abbreviations and acronyms the first time they are used.
3. Line 20-23: The sentence is too long and unclear.
4. Line 34-73: So many unnecessary details.
5. Line 74-86: The research advances about OFPs evolution in related species?
6. Line 103-106: Move these sentences to "OFP gene expression analysis".
7. Line 176: rename? What's the original names?
8. Line 194: semblable --> similar
9. Line 199: Wordy.
10. Line 270 and 294: Additionally --> In addition
11. Line 270-272: All the subcellular locations were predicted. Besides, this sentence is too long and unclear.
12. Line 278-280: Wordy.

Reviewer 2 ·

Basic reporting

The English language is not great, with some unusual choice of wording and with a few typographical errors. For example, the term 'owned' was used in a very odd way perhaps 20 times in the manuscript. Possible the authors already had some general proof reading but not by a scientific proof-reader. I suggest they obtain some help from a scientific proof-reader service.

Literature was adequately cited mostly with two problems:
1) Some of the software used did not have any citation information (e.g. MEGA7) and similarly for some equipment (e.g. Qubit 2.0)
2) The authors wandered into discussion in the Results section and cited too much literature, which was more appropriate to the Discussion section.

The structure, figures and tables were good. My only criticism is that figures and tables did not provide footnotes or titles that explained abbreviations or specific terms. This prevented the tables and figures from being able to stand alone from the main text.

Experimental design

The process of identifying OFP genes was rigorous and useful. I am not aware of a similar analysis of this kind previously made in Brassica rapa.

I was not as happy about the way the RNAseq analysis was performed. Many genes are under diurnal control and so it is important to sample tissues at the same time for the parents and hybrid.

Equally importantly, it is unclear exactly what plants were sampled, only that there were 27 in total and 3 had biological replicates. This has to be clarified.

No measurements of biomass / other plant yield parameters were made and so no inferences can possibly made about heterosis. It is unacceptable that the term heterosis appears in the title, and I think it is doubtful that it can even be included in the abstract, only the Discussion.

Validity of the findings

The gene copy descriptions are valid and useful. Most relevant underlying data were provided. FPKM values in each plant should be provided along with standard errors or other statistical measure of variation. It is unknown if the gene expression study had adequate controls as it was unclear about the nature of the replication. The discussion of heterosis is baseless without measurement of phenotype (e.g. biomass).

Additional comments

Give meaning of OFP in the abstract.
L65 Remove 'of cotton' (redundant)
L80 Replace 'fractionized' with 'fractionated'
L81 Restructure sentence as B. rapa has long season types (e.g. turnip) and a similar degree of morphological diversity as B. oleracea
L124 Rephrase this confusing sentence
L199-200 Note repeat information is provided in L204-205
L214 Replace ‘purifying’ with ‘purifying selection’
L222-223 This kind of statement belongs in the Discussion, not the Results.
L247 ‘which may also mean the functional diversgence of BraOFP7 gene’ is speculation, which does not belong in Results, which is for stating facts and observations. The same is true for the following complicated sentence (L248-252).
L252: The authors should spell out the evidence that indicates that introns were gained rather than lost in Brassica evolution.
L260 Explain the term ‘pI’
L281-282 Replace ‘it may become’ with ‘it is’
L303: It is unclear to me from your results that AT/Br pairs show a 1:1 relationship. As you covered previously, there is evidence for triplication followed by gene loss. Modify this statement.
L335-341: These references belong either in the Introduction or Discussion (or both), NOT in the Results.
L371 Cite your Figures or Tables in the Discussion section to support your arguments.
L417 Replace ‘transposable’ with ‘transposable element’ or ‘transposon’.
L468 This is complete speculation with no evidence provided to validate this discussion item. Phenotype information must be added or this aspect of the paper removed except for a sentence or two at the very most.

---

## Round 0.2 · accepted · Accept

The manuscript is improved and I am interested to accept it for publication.